

# Probing dielectric breakdown in Mott insulators through current oscillations

Joan Triadú-Galí[1,2,3⋆], Artur Garcia-Saez[1,4], Bruno Juliá-Díaz[2,3] and Axel Pérez-Obiol[5]

**1** Barcelona Supercomputing Center, BSC, 08034 Barcelona, Spain
**2** Institut de Ciències del Cosmos, Universitat de Barcelona, ICCUB, 08028 Barcelona, Spain
**3** Departament de Física Quàntica i Astrofísica, Facultat de Física, Universitat de Barcelona, 08028 Barcelona, Spain
**4** Qilimanjaro Quantum Tech, 08019 Barcelona, Spain
**5** Departament de Física, Universitat Autònoma de Barcelona, 08193 Bellaterra, Spain

⋆ joan.triadu@bsc.es

## Abstract

We investigate the dielectric breakdown of mesoscopic Mott insulators, a phenomenon where a strong electric field destabilizes the insulating state, resulting in a transition to a metallic phase. Using the Landau-Zener formalism, which models the excitation of a two-level system, we derive a theoretical expression for the threshold value of the field. To validate our predictions, we present an efficient protocol for estimating the charge gap and threshold field via non-equilibrium current oscillations, overcoming the computational limitations of exact diagonalization. Our simulations demonstrate the accuracy of our theoretical formula for systems with small gaps. Moreover, our findings are directly testable in ultracold atomic experiments with ring geometries and artificial gauge fields, as our method uses measurable quantities and relies on already available technologies. This work aims to bridge the gap between theoretical models and experimentally realizable protocols, providing tools to explore non-equilibrium mesoscopic phenomena in strongly correlated quantum systems.

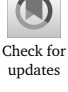

# 1 Introduction

Strongly correlated many-body quantum systems are a fundamental branch of condensed matter physics. The interaction between particles gives rise to remarkable phenomena such as superfluidity, topological order, and the separation of spin and charge dynamics [1–7]. These properties can be investigated across diverse experimental platforms, including ultracold gases in optical lattices [8–12], Rydberg atoms [13–17], and optical tweezer arrays [18, 19].

From the theoretical perspective, one of the simplest models that captures the nature of these systems is the Fermi-Hubbard model. Although it has only a few parameters, namely the filling, the tunnelling amplitude, and the on-site interaction between electrons, it has a rich phase diagram [20–22]. Part of the interesting phenomena exhibited by the Fermi-Hubbard model occurs only at the mesoscopic scale [23, 24]. The most archetypal one is the persistent current, a non-decaying charge current present when the system is at equilibrium and subjected to periodic boundary conditions [25–34]. The abnormal part of this current is that its existence is independent of any electromotive force applied to the lattice. This phenomenon is an example of the Aharonov-Bohm effect [35], which fixes twisted boundary conditions to the magnetic flux. Persistent currents have been widely observed in normal metals [36–41] and in recent experiments of ultracold fermionic atoms [42, 43].

Away from equilibrium, the charge current in mesoscopic Hubbard lattices has also attracted the attention of the scientific community, especially at half-filling [44–48]. In this regime, the 1D Fermi-Hubbard model exhibits a Mott-insulating ground state for repulsive on-site interactions [49], where the flow of electrons is suppressed because the opening of a charge gap ($\Delta$) between the lowest and upper Hubbard bands. However, when sufficiently strong electric fields are applied to the system, the electrons unfreeze their motion via a many-body Landau-Zener tunnelling and reach a metallic state. This phenomenon is known as the dielectric breakdown of Mott insulators [4, 46–48, 50–63]. Oka, Aoki, and Arita [46, 50] first studied this transition in mesoscopic lattices, proposing that the tunnelling follows the Landau-Zener formalism [64, 65], and predicting a quadratic dependence of the threshold field on the charge gap. More recently, numerical simulations have been performed to study the transition in spin-polarized rings [53] and in more realistic settings, with electrodes at the boundaries of mesoscopic chains [47, 48]. However, up to this day, the dielectric breakdown of mesoscopic Mott insulators lacks a robust theoretical expression that estimates the threshold field and agrees with their results.

In this work, we propose a theoretical formula for the threshold field by drawing a direct analogy between the Fermi-Hubbard and Landau-Zener spectra. To test our prediction, we design a protocol that can be implemented on digital quantum simulation platforms and ultracold atoms experiments. Our method consists in applying a uniform electric field to the ring and switching it off in the avoided crossing between the Hubbard bands. Following this

perturbation, the system exhibits non-equilibrium current oscillations which contain crucial information about the insulator-to-metal transition. Specifically, the oscillation frequency corresponds to the many-body charge gap, while the dependence of the oscillation amplitude on the applied field allows us to estimate the threshold field, $F_{th}$, for different interaction strengths and lattice sizes. Combining both quantities, $\Delta$ and $F_{th}$, we scatter the threshold field in terms of the gap $F_{th}(\Delta)$ and compare it with our proposed formula.

Advances in optical lattice and ultracold atom platforms open the possibility for direct experimental validation of our findings in controlled laboratory settings. Ring-shaped optical lattices have been successfully implemented [66,67], and the application of an artificial magnetic flux is feasible by introducing a complex phase factor to the tunnelling amplitude [68–70]. In addition, these technologies can be combined with quantum gas microscopy, a novel tool that enables single-site resolution and, by allowing precise potential shaping, helps achieve the low temperatures needed to observe long-range order in fermionic lattices [71–76]. These developments make our protocol of strongly correlated electrons out of equilibrium experimentally realizable in the near future.

This paper is structured as follows: In Sec. 2 we review the behaviour of the one-dimensional Fermi-Hubbard model under a static and linearly increasing magnetic flux. Then, in Sec. 3, the dielectric breakdown of Mott insulators is explained. In Sec. 4 we present the derivation of our theoretical formula. Afterwards, in Sec. 5, we detail the protocol to obtain the non-equilibrium current oscillations and how we calculate the threshold field from them. Finally, in Section 6, we present the results obtained and in Section 7 we discuss the findings of this study.

## 2 Fermi-Hubbard model under an electromagnetic field

The Fermi-Hubbard model describes correlated electrons in a lattice. The Hamiltonian of a ring of $L$ sites under an electromagnetic field reads

$$
H = -\tau \left[ \sum_{j,\sigma} e^{i\phi} c_{j+1,\sigma}^{\dagger} c_{j,\sigma} + h.c. \right] + U \sum_{j} n_{j\uparrow} n_{j\downarrow}, \tag{1}
$$

with periodic boundary conditions, where $c_{j,\sigma}^{\dagger}$ ($c_{j,\sigma}$) is the creation (annihilation) operator at site $j \in [1, L]$ and spin $\sigma = \uparrow, \downarrow$. The hopping integral $\tau$ accounts for the kinetic energy of the electrons and $U$ is the on-site Coulombian interaction. We fix $\tau = 1$, which defines the units of energy. We focus on the half-filled case, i.e. one particle per site ($n_{\uparrow} = n_{\downarrow} = L/2$). The electromagnetic field is introduced to the Hamiltonian with the Peierls phase $\phi = \frac{e}{c\hbar} \int_{j}^{j+1} \vec{A} \cdot \vec{dl}$ being $\vec{A}$ the magnetic vector potential. For a magnetic flux $\Phi$ piercing the ring, the Peierls phase evaluates to

$$
\phi = \frac{\Phi}{L}. \tag{2}
$$

We take $c = \hbar = e = R = 1$, being $R$ the radius of the ring. If the magnetic flux is not stationary but changes linearly over time, an electric field circulating throughout the ring appears via Faraday's Law $-d\Phi/dt = \int E \cdot dl$ and one can rewrite the time-dependent Peierls phase in terms of the electric field as

$$
\phi = a \cdot E \cdot t, \tag{3}
$$

where $a = \frac{2\pi}{L}$ is the lattice parameter, and $E$ is the electric field. $E$ has absorbed a minus sign because we are only interested in its absolute value. Note that the electric field is often written as $F = a \cdot E$ in the literature to avoid $L$ dependence in the Peierls phase. We use both forms in this work for convenience.

## 2.1 Constant magnetic flux

We start by discussing the equilibrium properties of the system when subjected to a constant magnetic flux ($\Phi = ct$), in both non-interacting and interacting regimes.

In the non-interacting limit ($U = 0$) the Hamiltonian is diagonal in quasimomentum space, and the eigenenergies are

$$\mathcal{E} = -2\sum_{k,\sigma} \cos\left[\frac{2\pi}{L}\left(k - \frac{\Phi}{2\pi}\right)\right], \tag{4}$$

where the sum in $k$ has $L/2$ different integers $\in [-L/2, L/2 - 1]$. Each pair of values $k$, $\sigma$ corresponds to a single-fermion energy level. In the absence of magnetic flux, the ground state energy corresponds to the $L$ levels with the lowest energy occupied, meaning the $L/2$ integers $k$ with the lowest absolute value for each $\sigma$.

When adding a magnetic flux the many-body spectrum has two different periodicities. First, a short period $\Phi = 2\pi$, known as the Aharonov-Bohm period, caused by the twisted boundary conditions of the magnetic flux. This pattern can be seen in Figure 1a) where there is a level crossing with the corresponding periodicity. Secondly, a longer period $\Phi = 2\pi L$, which is the amount of magnetic flux that the ground state takes to go up to the top of the Hubbard band and return to where it was. This one is the Bloch period, a consequence of the periodicity of the Peierls phase with the magnetic flux.

Another feature of the energy spectrum is the general shift of $\Delta\Phi = \pi$ between $L/2$ odd and $L/2$ even. For $L/2$ odd, the low energy level crossings appear at $\Phi = \pi(2n + 1)$, while for $L/2$ even the crossings are at $\Phi = 2\pi n$, with $n$ an integer. In this work we consider $L/2$ odd unless otherwise noted.

Apart from the effects on the energy levels structure, the application of the magnetic flux to a Fermi-Hubbard ring has further consequences. One of the most striking phenomena is the persistent current, a non-dissipative charge current present at equilibrium. To understand its nature it is convenient to define the current operator

$$I = -\frac{\partial H}{\partial \Phi} = \frac{i}{L}\sum_{j,\sigma}\left[e^{i\frac{\Phi}{L}}c_{j+1,\sigma}^{\dagger}c_{j,\sigma} - h.c.\right]. \tag{5}$$

Its expected value for eigenstates of the Hamiltonian with definite energy $\mathcal{E}$ is [77]

$$\langle I \rangle = -\frac{\partial \mathcal{E}}{\partial \Phi}. \tag{6}$$

The latter expression relates the current and energy spectra, implying that a parabolic behaviour in energy (as in Eq. (4)) corresponds to a linear behaviour in current even if no electromotive forces are involved. The current of the non-interacting ground state is displayed in Figure 1b), with its characteristic sawtooth shape following the Aharonov-Bohm period. In the thermodynamic limit, ($L \to \infty$), the energy spectrum flattens, implying the suppression of the persistent current. That is why the persistent current is considered a purely mesoscopic effect.

In the interacting regime ($U > 0$) the Hamiltonian is no longer diagonal in quasimomentum space. However, we can numerically diagonalize the Hamiltonian for small lattice sizes. In Figure 2a) we plot the low part of the spectrum for a finite interaction ($U = 1$). A distinguishing feature is the absence of level crossings in the low energy region, or in other words, the opening of a gap between the ground and excited energy levels, turning the level crossings into avoided crossings. The on-site interaction strength $U$ directly correlates with the size of the charge gap ($\Delta$), with stronger interactions leading to a larger gap. The gapped spectrum leads to a smooth, continuous current $I(\Phi)$ that still shows the Aharonov-Bohm periodicity, see Figure 2b).

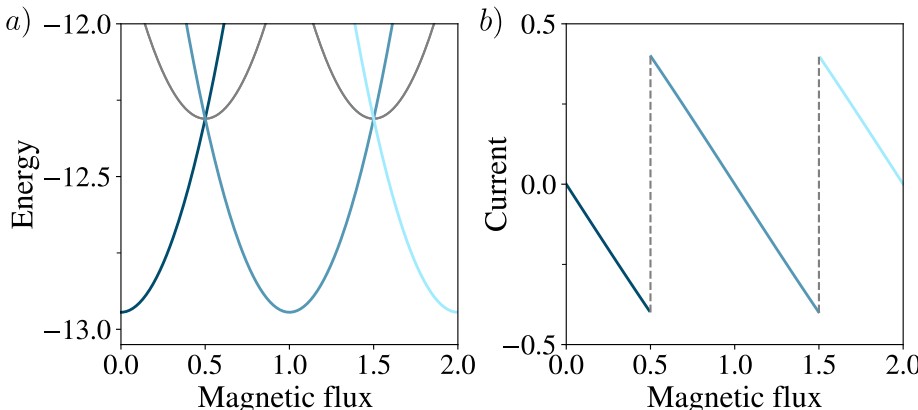

Figure 1: a) Non-interacting ($U = 0$) spectrum and b) persistent current for a half-filled $L = 10$ ring, obtained with Eqs. (4) and (6). Note the characteristic Aharonov-Bohm periodicity. Magnetic flux is in units of $\Phi_0/2\pi$, being $\Phi_0 = h/e$.

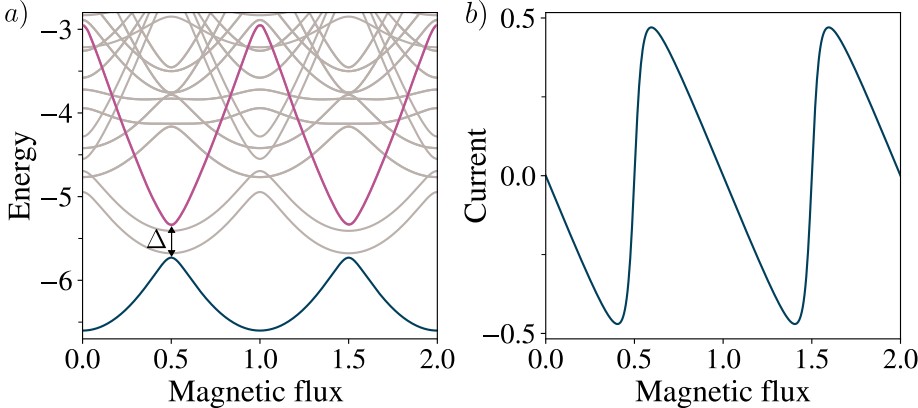

Figure 2: a) Spectrum and b) persistent current of the Fermi-Hubbard model with $U = 1$ for a half-filled $L = 6$ ring. Data obtained with exact diagonalization. In a), the charge excited state (colored in purple) is separated by $\Delta$ from the ground state at the avoided crossing. Magnetic flux is in units of $\Phi_0/2\pi$.

## 2.2 Linearly increasing magnetic flux

Away from equilibrium, when the magnetic flux is varied with a finite velocity ($\Phi = 2\pi E t$) the current displays a different pattern. We study this effect in ground states prepared with exact diagonalization in the absence of magnetic flux ($t = 0$), in the non-interacting ($U = 0$) and interacting ($U = 2$) regimes. For $t > 0$ we evolve the ground states under a linearly increasing magnetic flux, by numerically solving the time-dependent Schrödinger equation

$$|\Psi(t + \delta t)\rangle = e^{-iH\delta t}|\Psi(t)\rangle, \tag{7}$$

with $\delta t = 0.05$, using the *Openfermion* library [78]. We follow the same protocol for the rest of the simulations in the work.

The non-equilibrium current is shown in Figure 3 with solid lines, while the persistent current is plotted in dotted lines. For $U = 0$ the time-evolved wave function is at all times fully populating the same instantaneous eigenstate. As the eigenstate does not correspond, in general, to the ground state, the current does not show the Aharonov-Bohm period and shows no longer a sawtooth shape. In this case, it oscillates with the Bloch period and has higher amplitude than in equilibrium, as shown in Figure 3a). For $U > 0$, the situation is different.

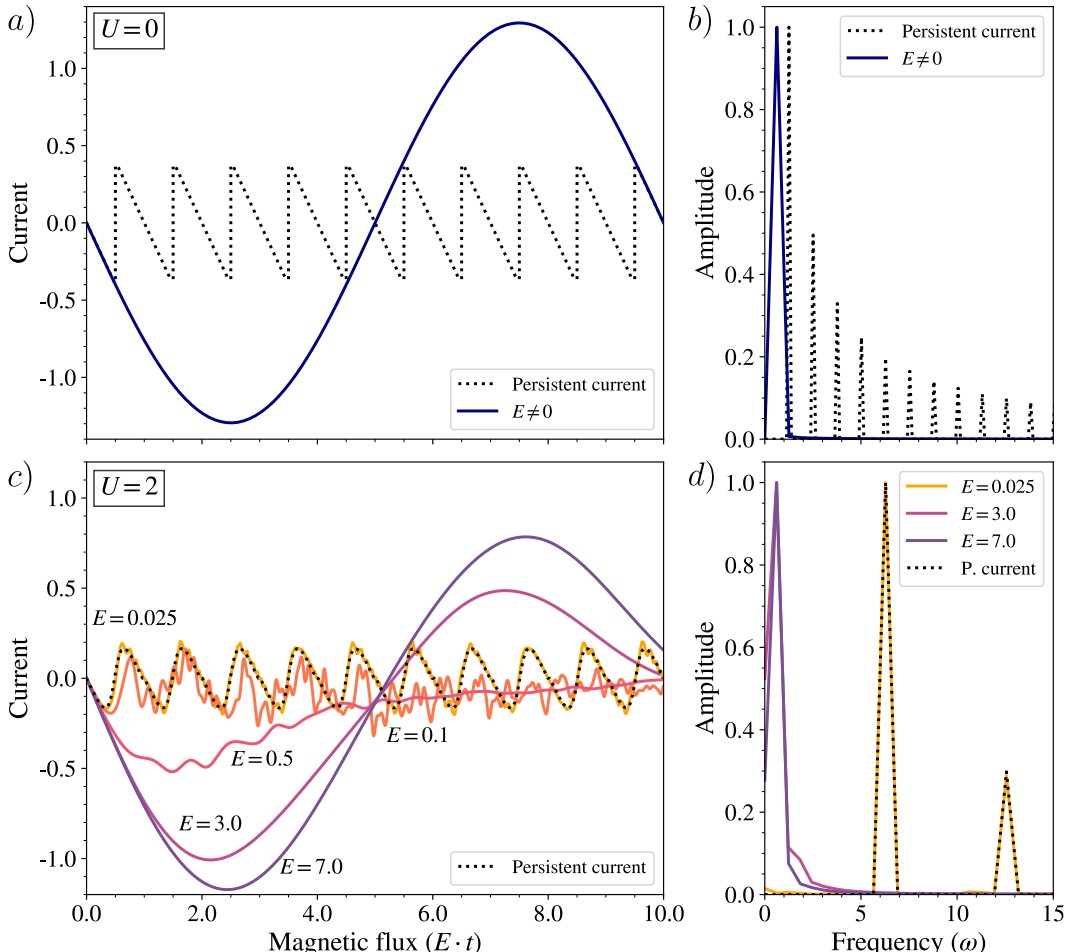

Figure 3: Left column: Charge current of $L = 10$ half-filled rings in the a) non-interacting ($U = 0$) and c) interacting ($U = 2$) regimes. Different curves correspond to different fields. The persistent current is displayed with dotted lines. A similar plot of the interacting regime can be found in [46]. Right column: b), d) Fourier transform of a) and c), respectively. In d), current curves without a clear frequency have been omitted for clarity.

The existence of the charge gap allows the wave function to remain in the insulating state, as long as the evolution is adiabatic. That is why for sufficiently small fields the current is the same as in equilibrium, as shown in Figure 3c). For stronger electric fields, tunnelling to the excited spectrum becomes relevant, and the persistent current pattern is lost around the first avoided crossing, at $E \cdot t = 0.5$. This is a signature of the dielectric breakdown of the system. It is relevant to note that the system does not necessarily excite to the first excited state, since there are excitations that do not couple to the electric field. For even larger fields, a Bloch oscillation pattern is observed, since the wave function can cross multiple avoided crossings and reach the top of the band, thus recovering the pattern for $U = 0$, as reported in previous studies [46]. In Figure 3d) one can see how the frequency of the oscillations changes as the field increases: from the persistent current pattern (small period, high frequency) to the Bloch oscillation regime (large period, small frequency). The objective of this study is to characterize this dielectric breakdown in terms of the electric field using the Landau-Zener formalism.

## 3 Dielectric breakdown of Mott insulators

The dielectric breakdown of Mott insulators occurs when a strong electric field destabilizes the insulating state, leading to a transition to a metallic phase. The mechanism behind this breakdown is analogous to the Schwinger effect in quantum electrodynamics, where strong fields induce electron-positron pair production in vacuum [52, 54, 79]. In Mott insulators, the electric field triggers the creation of doublon-hole pairs which act as mobile charge carriers and drive the system out of the insulating state.

The asymptotic probability of excitation away from the insulating state is

$$P = \exp\left(-\pi \frac{F_{th}}{F}\right), \tag{8}$$

which defines the threshold field $F_{th}$ of the transition. Oka, Aoki and Arita [46, 50] proposed that this breakdown is governed by Landau-Zener tunnelling, confirming that the breakdown threshold is tied to the Mott gap, with a critical field $F_{th} \propto \Delta^2$ for small lattice sizes and gaps. In their later works, Oka et al. [51, 52] established the analogy between the dielectric breakdown of Mott insulators with the Schwinger effect and computed the threshold field in the thermodynamic limit (TL) by applying the Dykhne-Davis-Pechukas (DDP) method [80–82]. The resulting formula is $F_{th}^{TL} \simeq \Delta/2\xi$ where $\xi(U)$ is the correlation length of the system. This quantity was originally found by Stafford and Millis [83, 84]

$$1/\xi(U) = \frac{4}{U} \int_1^\infty dy \frac{\ln(y + \sqrt{y^2 - 1})}{\cosh(2\pi y/U)}. \tag{9}$$

It is defined as the parameter that describes the exponential decay of the Drude weight $D(L)$ of the system for growing $L$

$$D(L) = \frac{1}{2} \left.\frac{d^2 \mathcal{E}_0}{d\Phi^2}\right|_{\Phi=0} \sim \exp(-L/\xi), \tag{10}$$

being $\mathcal{E}_0$ the ground state energy. Additionally, the correlation length can be interpreted as the typical separation between the doublon and hole quasiparticles [83–85].

In the next section, we propose a quantitative formula for the threshold field of mesoscopic systems based on the original work from Zener [64]. We make a direct analogy between his formalism and the Fermi-Hubbard model, by making an explicit connection between the two spectra.

## 4 Threshold field in mesoscopic Mott insulators

The Landau-Zener (LZ) model (see [86] for a review) consists of a two-level system described by the Hamiltonian

$$H^{LZ} = \begin{pmatrix} vt & \Delta E/2 \\ \Delta E/2 & -vt \end{pmatrix}, \tag{11}$$

that has two adiabatic eigenstates $|0\rangle, |1\rangle$ with their corresponding energies, $E_0(t), E_1(t)$ plotted in Figure 4. The two levels are separated by an energy gap which is minimal in the avoided crossing, at time $t = 0$, where it has a value $\Delta E$. $v$ is the linear bias applied to the system. In the LZ Hamiltonian, the gap opens following the asymptotes $\epsilon_\pm(t) = \pm vt$, which correspond to the $\Delta E = 0$ energy levels (Figure 4) and fulfill

$$v = \frac{1}{2} \left| \frac{d(\epsilon_+(t) - \epsilon_-(t))}{dt} \right|. \tag{12}$$

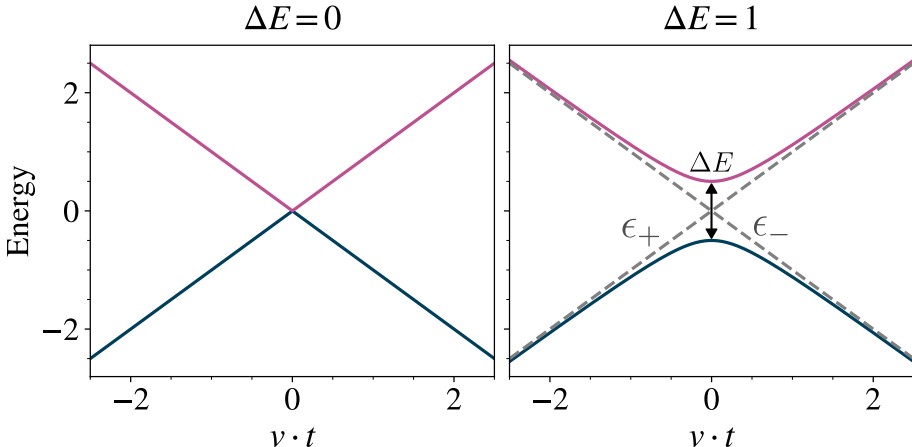

Figure 4: Spectrum of the Landau-Zener Hamiltonian (Eq. (11)) for $v = 1$. The plot on the right shows the gap at the avoided crossing, $\Delta E$, and $\epsilon_\pm(t) = \pm vt$ in dashed lines, i.e., the energy levels for $\Delta E = 0$.

Zener [64] found that when evolving the system from $t = -\infty$ to $t = \infty$, the probability of measuring the wave function in the excited state is

$$P = \exp\left(-\pi \frac{\Delta E^2}{2|\frac{d}{dt}(\epsilon_+ - \epsilon_-)|}\right). \tag{13}$$

We apply this formalism in our case of study, the 1D Fermi-Hubbard model, to find an expression for the threshold field of the system, $F_{th}$. Although the Hamiltonian is periodic, around the crossing ($\Phi = \pi$) the system can be approximated with this model if the transition region is small enough, that is, for small gaps ($\Delta \approx 1$) [46,50,64]. We are also assuming that only the ground state and a single excited state contribute to the dynamics of the many-body system.

To use the LZ approximation on the Fermi-Hubbard model, we match the LZ excitation probability, Eq. (13), to the Fermi-Hubbard asymptotic excitation probability, Eq. (8). First, we identify the LZ gap $\Delta E$ with the many-body charge gap $\Delta$. Next, we assume that the asymptotes $\epsilon_\pm$ are the first two Fermi-Hubbard energy levels for $U = 0$, namely the ground state and the ground state shifted by $k \rightarrow k + 1$. This assumption allows us to compute $\frac{d}{dt}(\epsilon_+ - \epsilon_-)$, the other term in Eq. (13). We use Eq. (4) to find the corresponding curves and evaluate the derivative at the crossing

$$\left.\frac{d}{dt}(\epsilon_+ - \epsilon_-)\right|_{\Phi=\pi} = \left.\frac{d}{d\Phi}(\epsilon_+ - \epsilon_-)\right|_{\Phi=\pi}\frac{d\Phi}{dt} = \frac{8}{L} \cdot 2\pi E. \tag{14}$$

After identifying terms, the resulting expression for the excitation probability of the Fermi-Hubbard model reads

$$P = \exp\left(-\pi \frac{\Delta^2 \cdot L}{32\pi E}\right), \tag{15}$$

where the threshold field is

$$E_{th} = \frac{\Delta^2 \cdot L}{32\pi}. \tag{16}$$

This expression suggests that the transition does not take place for large systems, as the $E_{th} \propto L$. However, in the Peierls phase, the field is multiplied by the lattice constant, which

is inversely proportional to the system size, and cancels out the factor $L$ (Eq. (3)). It is then convenient to write the threshold field in units of $F$

$$F_{th} = \frac{\Delta^2}{16}. \tag{17}$$

In these units, the threshold field is independent of $L$. We expect this equation to be reliable as long as the two-level approximation holds. However, when the work of the applied field becomes comparable to the gap of the next excited state, i.e., $2\pi E_{th} = LF_{th} \approx 2\Delta$, another state contributes to the wave function and the formula is no longer valid. In the lattice sizes considered in this work, this happens for $U \approx 5$. For lattice sizes up to $L \approx 10^2$, the two-level approximation remains valid for interaction strengths $0 < U \approx 3$. In the thermodynamic limit, the spectrum of the Fermi-Hubbard model becomes flat and the LZ formalism is no longer applicable as $v \to 0$, so the formula is invalid regardless of the magnitude of the field.

An important feature of Eq. (17) is that it is in accordance with the finite-size effects of the model, which require the threshold field at mesoscopic scale to be smaller than in the thermodynamic limit. The reason is that the Drude weight in finite lattices has a non-vanishing value for finite interaction strength, see Eq. (10), while is totally suppressed in an infinite chain. In other words, the effective mass of the carriers is smaller at mesoscopic scale than at the bulk limit. This implies that the critical field to break the insulating phase in a finite lattice should be smaller than in an infinite one. By expanding the correlation length of the system for small $U$ [52, 83, 84], one obtains that the threshold field at the bulk limit is

$$F_{th}^{TL} \approx \frac{\Delta^2}{8}, \tag{18}$$

for $\Delta \to 0$, which is greater by a factor of two than our theoretical formula, Eq. (17), as desired. In the next section, we further validate our theoretical expression by performing numerical simulations to confirm its applicability.

## 5 Numerical simulations

To verify the accuracy of our prediction one needs: 1), to find a way of driving the system out of equilibrium without involving higher excited states and, 2), to describe its behaviour in terms of the threshold field. These tasks are carried out in the following two subsections.

### 5.1 Non-equilibrium current oscillations

In order to excite the system to only a single state it is necessary to evolve the system no further than the first avoided crossing. Afterwards, the fraction of the wave function that has leaked to the excited state can leak again to the upper zones of the spectrum by transitions that are not within the scope of our theoretical formula. Then, to numerically study the dielectric breakdown we apply a linearly growing magnetic flux with a certain velocity from the ground state of $\Phi = 0$ ($t = 0$) until we reach the avoided crossing at $\Phi = \pi$ ($t = 1/(2E)$). Once there, we maintain the magnetic flux constant (suppressing the electric field) and let the system evolve. Under the two-level approximation, the instantaneous wave function for $t = 1/(2E)$ is

$$|\Psi\rangle = c_0 |0\rangle + c_1 e^{i\alpha} |1\rangle, \tag{19}$$

where $c_0, c_1, \alpha \in \mathbb{R}$ and $|0\rangle$, $|1\rangle$ are the instantaneous (or adiabatic) eigenstates at $\Phi = \pi$. Correspondingly, the expected value of the current from this time onward is

$$\langle I \rangle = c_0^2 \langle I_0 \rangle + c_1^2 \langle I_1 \rangle + A\cos(\alpha + \phi - \Delta \cdot t). \tag{20}$$

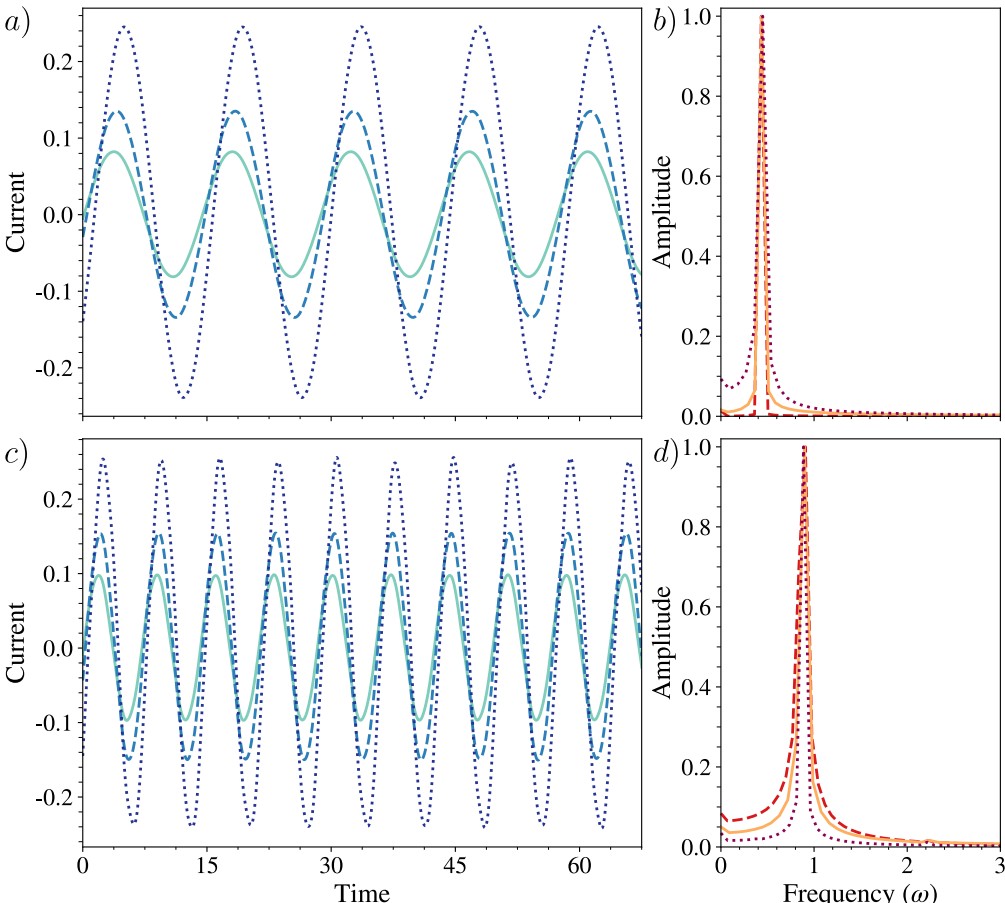

Figure 5: Left column: current response after switching off the field at the avoided crossing ($\Phi = \pi$, $t = 1/2E$) for different on-site interactions. Different curves correspond to different fields. a) $U = 1.5$. Solid line is $E = 0.0078$, dashed $E = 0.013$, and dotted $E = 0.032$. c) $U = 2.5$. Solid line is $E = 0.045$, dashed $E = 0.075$, and dotted $E = 0.19$. Right column: b), d) Fourier transform of a) and c), respectively. The amplitudes are normalized by the peak. Line style of each field is conserved.

From this expression one can see that the current oscillates with a frequency equal to the charge gap between the two adiabatic levels, $\Delta$. Conveniently, this is a relevant magnitude of our theoretical formula. $A$ is the amplitude of the oscillations

$$A = 2c_0 c_1 |\langle 0|I|1\rangle| \propto \sqrt{P_+(1-P_+)}, \tag{21}$$

which, assuming normalized coefficients, $c_0^2 + c_1^2 = 1$, is expressed as a function of the probability to measure the adiabatic excited state, $P_+ = c_1^2$.

Furthermore, since the adiabatic ground (excited) state is at a maximum (minimum) of energy, the non-oscillatory terms in Eq. (20) must vanish according to Eq. (6), which causes the current to oscillate around zero. In the left column of Figure 5, the current of the system after switching off the field ($t > 1/2E$) shows an oscillatory pattern with a variable period and amplitude. As expected, the period depends only on the on-site interaction, whereas the amplitude of these oscillations grows with the field applied, evidencing a clear relation between the field and the movement of the electrons. Also, the oscillations are centered around zero as predicted. These findings strongly indicate that the two-level approximation is valid, and we also confirmed that the contribution of higher excited states is negligible using exact diagonalization.

These well-behaved oscillations allow us to obtain crucial information about equilibrium and non-equilibrium properties of the ring by studying their frequency and amplitude. The frequency equals the charge gap $\Delta$ of the spectrum, and to extract its value we compute the Fourier transform of the current after the field is switched off, see the right column of Figure 5. For each interaction, which implies a particular energy gap and a single current frequency, we simulate a range of applied electric fields and average the set of obtained frequencies to gain statistical significance. The amplitude, which is field-dependent, contains information on the threshold field of the system, see next subsection. To obtain its magnitude, we take the maximum and minimum values of the current for each oscillation and do the average for all the oscillations obtained under the same conditions.

## 5.2   Finite coupling Landau-Zener

We relate the amplitude of the current oscillations and the threshold field using an exact time integration of the LZ model made by Vitanov et al. [87,88]. Using their work, one can relate the probability of excitation at the avoided crossing $P_+$ with the threshold field, when starting the evolution at $t = -\infty$, as

$$P_+ = \frac{\pi}{8} \exp\left(\frac{-\pi\omega^2}{4}\right) \left| \frac{2}{\Gamma\left(\frac{1}{2} - \frac{i\omega^2}{4}\right)} + \frac{(-1+i)}{\sqrt{2}} \frac{\omega}{\Gamma\left(1 - \frac{i\omega^2}{4}\right)} \right|^2, \tag{22}$$

where $\omega^2 = E_{th}/E$. Since we measure the amplitude of the current oscillations we are not interested in $P_+$ but in $P_+(1 - P_+)$, see Eq. (21). The function resulting from this product has a smooth behaviour in the quasi-adiabatic regime we are working on, $\sqrt{2/3} \le \omega \le \sqrt{2}$, that allows us to do an expansion around $\omega = 0$

$$\ln\left(P_+(1 - P_+)\right) \approx -\ln(4) - \frac{\pi\omega^2}{2} + \frac{\pi\omega^4}{8}(\pi - 4\ln(2)) + \mathcal{O}(\omega^8), \tag{23}$$

with negligible error in the aforementioned regime. Since $A \propto \sqrt{P_+(1 - P_+)}$, we relate the amplitude of the oscillations to the threshold field

$$-\ln(A^2) \approx \frac{\pi}{2}\left(\frac{E_{th}}{E}\right) - \frac{\pi}{8}(\pi - 4\ln(2))\left(\frac{E_{th}}{E}\right)^2 + C, \tag{24}$$

$C$ being a constant. With this expression, we numerically estimate the threshold field, $E_{th}$. We simulate a range of electric fields $E$ and compute the corresponding amplitudes $A$. Then, performing a numerical fit to the obtained amplitudes as a function of $1/E$ using Eq. (24), we extract the two free parameters, $E_{th}$ and $C$.

We expect this approximation to be valid when 1) the two-level approximation holds, i.e. $2\pi E_{th} = LF_{th} < 2\Delta$ which is true for fields up to $E \approx 1$ with lattices sizes of this work, and 2) when the gap at the beginning of the evolution ($\Delta_0$) is much larger than the gap at the avoided crossing, $\Delta/\Delta_0 \ll 1$. This condition does not hold when $\Delta \gg 1$ because the spectrum flattens, faster for greater L (see Eq. (10)). This results in $\Delta \approx \Delta_0$ which makes Eq. (24) no longer applicable, regardless of the electric field applied.

# 6   Results

In this section we present the main results of the work. By extracting the frequency of the current after exciting the system with an electric field, we have obtained the charge gap, $\Delta$.

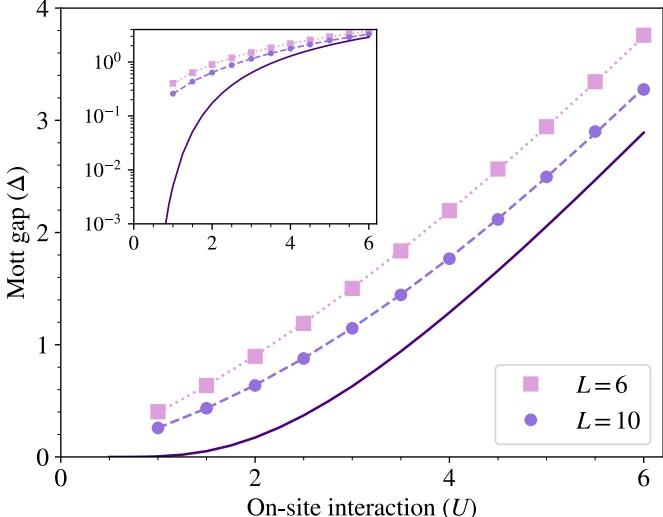

Figure 6: Many-body charge gap for different on-site interactions and lattice sizes. Markers represent values extracted via Fourier transform of non-equilibrium current oscillations. Dotted and dashed lines show the exact diagonalization results for $L = 6, 10$, respectively. Solid line shows the gap at thermodynamic limit. Inset shows logarithmic scale in the y-axis for clarity on finite-size effects close to $U = 0$.

Figure 6 shows $\Delta(U, L)$ for repulsive interactions $1 \leq U \leq 6$, and lattice sizes $L = 6, 10$. To test the accuracy of the proposed protocol, we have compared our results with the gaps obtained by exact diagonalization, shown in dotted and dashed lines in the figure, respectively. The gaps obtained with the current response are found to match with high fidelity the gaps from exact diagonalization. Together with these two lines, we show in Figure 6 the charge gap $\Delta(U)$ in the thermodynamic limit (solid line). The gap in the thermodynamic limit is significantly different than the gap for the lattice sizes we are studying, particularly near the critical point for the metal-insulator transition ($U = 0$) where the finite-size effects are stronger. This behaviour is shown more clearly in the logarithmic plot in the inset of Figure 6.

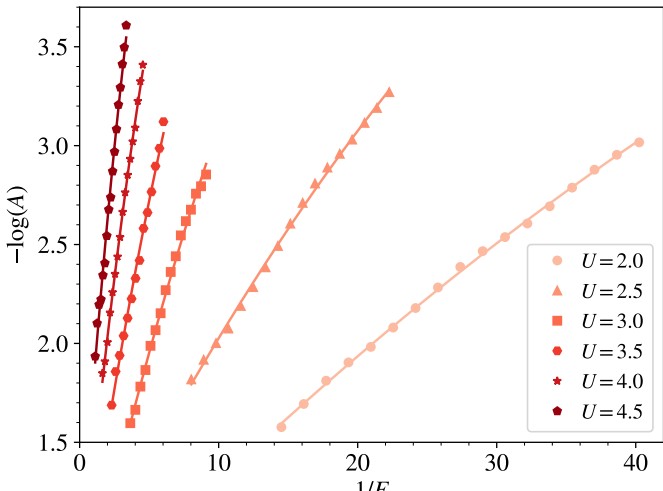

Figure 7: Amplitude of the current oscillations $A$ in terms of the inverse electric field $1/E$, for different interactions and $L = 10$. Markers are data and lines are fittings to Eq. (24).

We also extract the amplitudes of the current oscillations, $A$, by averaging the maximum and minimum values of the current for each period. This procedure allows us to test the behaviour of $A(1/E)$ characteristic of the LZ formalism. In Figure 7 we plot $-\ln(A^2)$ as a function of $1/E$ for $L = 10$ and various interactions. In the same figure, we plot for each interaction a fit to Eq. (24) where the threshold field $E_{th}$ is a free parameter. The fit is reasonably accurate, specially for the smaller interaction strengths.

With the previous fittings we obtain $E_{th}(U, L)$, and using $\Delta(U, L)$, we plot the threshold electric field $E_{th}(\Delta)$ in Figure 8, for $L = 6, 10$. In the same figure, we plot our formula of Eq. (16) for each lattice size, with dashed and dotted lines, respectively. The data points perfectly overlap with the theoretical curves for small gaps. For $L = 6$, the fields start to deviate at gaps $\Delta \gtrsim 2$, while for $L = 10$, the data points and the curve start to disagree at $\Delta \gtrsim 1$. This disagreement arises when the assumption that the minimum gap is much smaller than the initial gap does not hold. Furthermore, simulations fail earlier for $L = 10$ than $L = 6$ because the spectrum flattens faster for greater $L$, see Eq. (10). We have observed that the deviation of the data points in a particular gap $\Delta$ (and the corresponding $U$) in both lattice sizes coincides with the correlation length (which is a decreasing function of $U$) dropping below the lattice size. When $\xi < L$, the Drude weight starts to decay exponentially, implying a flattening of the spectrum and the breakdown of our approximation. In these conditions, the numerical fitting is no longer accurate, and the data deviates from the theoretical curve. To make this explicit, data points in this regime, $\xi < L - 1$, are plotted with empty markers.

In Figure 9 we plot the threshold field in units of $F$ for $L = 6, 8, 10$. We consider only the data points in the $\xi > L - 1$ regime. Our theoretical formula in Eq. (17) is plotted in dashed lines, and, for comparison, the threshold field in the thermodynamic limit is plotted in dotted lines. The data points fall on our proposed theoretical curve, confirming that the threshold field is smaller for mesoscopic systems than in the thermodynamic limit. Furthermore, we plot three different system sizes to highlight the independence of the threshold field ($F_{th}$) on the system size.

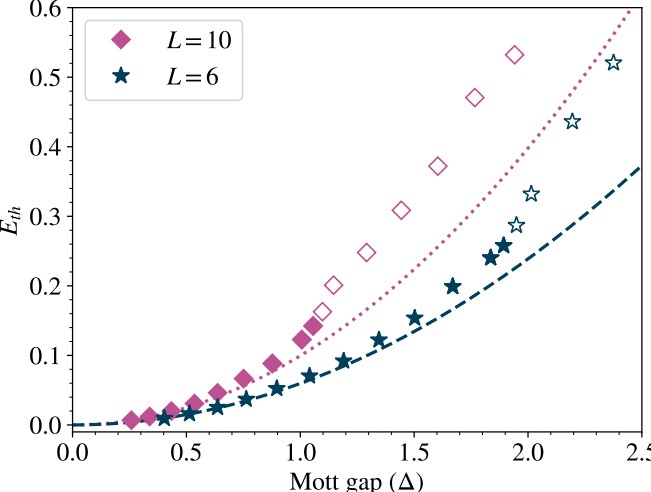

Figure 8: Threshold field $E_{th}(\Delta)$ for different number of sites. Dashed (dotted) line is Eq. (16) for $L = 6$ ($L = 10$). Empty markers show the $\xi < L - 1$ regime. $\xi$ is estimated using Eq. (9). Data for $L = 8$ is not shown for clarity but it is analogous to the other lattice sizes.

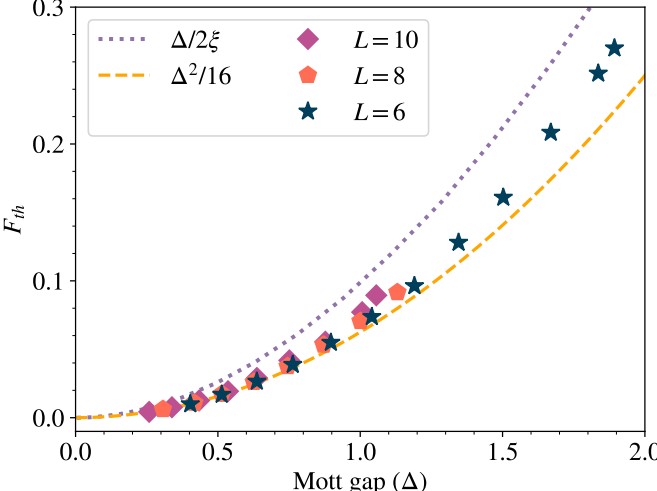

Figure 9: Threshold field $F_{th}(\Delta)$ for different ring sizes. Markers show the threshold field in the $\xi > L - 1$ regime. Dashed line is Eq. (17), dotted line is the threshold field at the thermodynamic limit ($F_{th}^{TL}$).

## 7  Discussion

We study the dielectric breakdown of the mesoscopic 1D Fermi-Hubbard model. We propose a formula for the threshold field within the Landau-Zener formalism, which depends quadratically on the charge gap and depends on the slope of the spectrum at the avoided crossing, analogously to single particle systems. Additionally, our formula is smaller than the threshold field at the bulk limit, in accordance to the finite-size effects of the charge stiffness. To check the validity of our prediction we numerically estimate the threshold field by studying the frequency and the amplitude of non-equilibrium current oscillations.

The frequency of the oscillations has the same value as the charge gap, which is a fundamental quantity to study the dielectric breakdown of Mott insulators. Although its value is exactly known for $L \to \infty$, for finite number of sites must be obtained by methods of exact diagonalization that scale exponentially with the size of the system. The advantage of our approach is that we accurately estimate this magnitude that depends on the Coulombian repulsion, $U$, and the number of sites, $L$, by computing the Fourier transform of the current response. With the amplitude of the oscillations, and following the work from Vitanov [87, 88], we estimate the threshold field for a given $U$ and $L$. As we know the charge gap for each of these parameters, we can plot $E_{th}(\Delta)$ and compare it to our proposed formula. The numerical results match our expression for small gaps, i.e. when $\xi \gtrsim L$. For larger gaps, equation (24) is no longer applicable, so the data points on this regime presented in this work are not significant to assert nor reject our theoretical prediction. Nevertheless, we note that results for systems with such large gaps, [48, 53], are compatible with our predictions.

Works studying the dielectric breakdown of mesoscopic Hubbard lattices often use the gap at the thermodynamic limit, because it is the only case where it is known for all $U$. However, the gap at the bulk limit is significantly different for a finite number of sites. With that flawed approximation, the behaviour of the threshold field in terms of the gap may be biased. In our work, we used a very accurate estimation of the real value of the charge gap, while avoiding exact diagonalization. This feature makes our protocol compatible with computational platforms that scale efficiently with system size, such as Tensor Networks and digital quantum computing. Additionally, our proposed method is also well-suited for current experimental platforms, as it relies on techniques that are already available in ultracold atom

and quantum simulation setups. The ability to engineer uniform electric fields and measure non-equilibrium currents with single-site resolution makes it possible to observe the predicted oscillatory dynamics and extract the threshold field. With this approach we also open the door to study a similar transition close to half-filling. In those conditions the spectrum is gapless in the low energy region, but the current also presents a breakdown-like pattern [46] and similar non-equilibrium current oscillations [45].

Finally, our expansion in Eq. (24) relating the probability of excitation at the avoided crossing with the threshold field can be extrapolated to other one or many-body systems where there is a two-level dynamical phase transition. By observing the amplitude of any oscillating observable at the avoided crossing, one can estimate the threshold field related to the transition.

## Acknowledgments

We thank Alessio Celi, Sergi Masot-Llima, Jofre Vallès-Muns and Piotr Sierant for useful discussions.

**Funding information** This work has been funded by Grant PID2023-147475NB-I00 funded by MICIU/AEI/10.13039/501100011033 and FEDER, UE, by grants 2021SGR01095 and 2021SGR00907 from Generalitat de Catalunya, and by Project CEX2019-000918-M of ICCUB (Unidad de Excelencia María de Maeztu). T-G. J. acknowledges funding from the Spanish Ministry for Digital Transformation and of Civil Service of the Spanish Government through the QUANTUM ENIA project call - Quantum Spain, EU through the Recovery, Transformation and Resilience Plan – NextGenerationEU within the framework of the Digital Spain 2026.

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
