# Peer review of "Probing dielectric breakdown in Mott insulators through current oscillations"

_SciPost Physics, doi:SciPost Phys. 19, 010 (2025)_

## Round 1 · Referee Report · Anonymous (Referee 1) · 2025-4-6

Report

This study investigates the threshold electric field strength at which dielectric breakdown occurs when a magnetic flux is applied to a Hubbard model on a ring, thereby generating an electric field along the ring. Employing the Landau-Zener model, the authors clarify the relationship between the charge gap and the breakdown threshold. Furthermore, they point out that the charge gap and the breakdown threshold can be estimated from the frequency and amplitude of the current induced by the flux, respectively. These two approaches to assess the threshold are numerically validated to demonstrate their reliability.

The novelty of this paper lies in its elucidation of how the threshold behavior is affected in finite systems, in contrast to previous studies that mainly focused on the one-dimensional Hubbard model in the thermodynamic limit. I believe that this study is suitable for publication. However, I request the authors to address the following comments.

Requested changes

1- Equation (17) relates the charge gap to the threshold field. Could the authors quantitatively clarify the parameter range (regarding the interaction strength $U$ and the system size $L$) over which this relation remains valid?

2- Equation (24) is an expansion in terms of $E_{th}/E$. Could the authors clarify the range of electric field strength $E$ over which this approximation is quantitatively reliable?

3- While the present study focuses on Mott insulators, the approach may be extendable to other gapped systems such as semiconductors. Could the authors comment on whether there are qualitative differences in the threshold behavior due to many-body effects that would not appear in single-particle gapped systems?

4- In Fig. 2(a), the part of the energy spectrum with $E \gtrsim -3$ appears to be omitted. Including the full spectrum within the displayed range might improve clarity unless there is a specific reason for excluding it.

Recommendation

Ask for minor revision

  • validity: high
  • significance: ok
  • originality: good
  • clarity: good
  • formatting: good
  • grammar: perfect

Author:  Joan Triadú  on 2025-05-26  [id 5519]

(in reply to Report 1 on 2025-04-06)

We thank the referee for the careful revision of our work. We have meticulously revised the points that were raised in the report and corrected them.

  • Equation (17) relates the charge gap to the threshold field. Could the authors quantitatively clarify the parameter range (regarding the interaction strength $U$and the system size $L$) over which this relation remains valid?

The Landau-Zener formalism is an adequate framework to describe the dielectric breakdown of mesoscopic Mott insulators for gaps up to the order of magnitude of the tunneling coefficient. The Landau-Zener formalism does not work beyond that range of parameters because to excite a system with a larger gap, one needs to apply a strong electric field which couples to more than one excited state, and one cannot use a two-level approximation anymore. We expect this approximation to fail when the work done by the electric field throughout the whole lattice is comparable to the energetic gap of the next excited state, that is, $2\pi E_{th}=LF_{th}\approx 2\Delta$ (using the same units convention that in the main text). In the lattice sizes we have studied, this happens for $E\approx1$, $U\approx5$. Based on this criterion, for lattices of size $L\approx10^2$ (the limit of mesoscopic scale), the two-level approximation remains valid for interaction strengths of $0<U\approx3$ and fields $E\approx10^{-2}$. Although $2\pi E_{th}=LF_{th}$ grows linearly with the system size, a finite Mott gap in such systems requires interaction strengths comparable to the tunnelling amplitude. This is consistent with the findings of Ref. [48], which reports results for $L= 300$ and $U=1.5$ that align with ours.

Changes: We have added this discussion, including the relevant range of interaction strength $U$ and the system size $L$ over which Eq. (17) is valid, to the main text, below Eq. (17).

  • Equation (24) is an expansion in terms of $E_{th}/E$. Could the authors clarify the range of electric field strength $E$ over which this approximation is quantitatively reliable?

Mathematically, $\ln\left(P_+(1-P_+)\right)$ has a maximum of $0.2\%$ relative error with respect to the exact expression in the range of $0<E_{th}/E<3$, see the attached file for more details. Despite this high accuracy, Eq. (24) can fail for the Fermi-Hubbard Model in two cases:

1) When the gap at the beginning of the evolution ($\Delta_0$) is comparable to the gap at the avoided crossing ($\Delta$), Eq. (24) becomes invalid regardless of the applied field. We find that this breakdown occurs when $L\approx\xi(U)$. In the lattice sizes of this work, this corresponds to an on-site interaction of $U\approx3.5$, so we can only confirm our predictions up to these interaction strengths using Eq. (24). For lattices up to $L\approx10^2$, $L\approx\xi(U)$ is met at interaction strengths $U\approx1$. Note that this condition does not speak about the validity of Eq. (17), but the suitability of using Eq. (24) to check the theoretical prediction.

2) When the work done by the electric field across the lattice becomes comparable to the energetic gap of the next excited state, i.e., when $2\pi E_{th}=LF_{th}\approx2\Delta$. This condition is met for $E\approx1$ in the system sizes of this work. For lattices up to $L\approx10^2$, the breakdown of the two-level approximation happens at $E\approx10^{-2}$.

For pure two-level systems that evolve from $t=-\infty$ (or equivalently $\Delta/\Delta_0\ll 1$ ), Eq. (24) is appropriate in the range $0<E_{th}/E<3$ no matter the strength of the field $E$.

Changes: We added the second condition (with the range of E for which the approximation is quantitatively reliable) after Eq. (24). The first condition was already mentioned there, but we now explicitly emphasize that when $\Delta\approx\Delta_0$, Eq. (24) fails regardless of the field strength. The breakdown of the approximation at $L\approx\xi(U)$ is discussed in the results section (Sec. 6).

  • While the present study focuses on Mott insulators, the approach may be extendable to other gapped systems such as semiconductors. Could the authors comment on whether there are qualitative differences in the threshold behavior due to many-body effects that would not appear in single-particle gapped systems?

The dielectric breakdown of Mott insulators shows a threshold behavior analogous to single-particle gapped systems. In fact, Zener’s original work Ref. [64], which underpins our theoretical approach for the Fermi-Hubbard model, was developed for single-particle gapped systems. That is why we don’t expect a single-particle gapped system to display qualitative differences in the threshold behavior in respect of our results as long as: -It effectively behaves as a two-level system -The diagonal term of the Hamiltonian varies linearly in time near the crossing -The off-diagonal term (coupling) is constant -The transition occurs around the crossing (the rest of the evolution is adiabatic)

Changes: We added a comment in the discussion: “We propose a formula for the threshold field within the Landau-Zener formalism, which depends quadratically on the charge gap and depends on the slope of the spectrum at the avoided crossing, analogously to single particle systems.”

  • In Fig. 2(a), the part of the energy spectrum with $E≳ −3$ appears to be omitted. Including the full spectrum within the displayed range might improve clarity unless there is a specific reason for excluding it.

Thank you for the suggestion of changing our figure, we have included this improvement in the new version of the manuscript.

Changes: Included the full spectrum within the displayed range in Figure 2(a).

Attachment:

Expansion_of_Eq___23_.pdf

---

## Round 1 · Referee Report · Anonymous (Referee 2) · 2025-4-20

Strengths

(1) The paper presents a systematic numerical study of dielectric breakdown in one-dimensional Mott insulators across a wide parameter range using the Fermi-Hubbard model.

(2) By applying the Landau-Zener formalism, the authors accurately extract excitation gaps from current oscillations, with results consistent with exact diagonalization.

(3) The work provides detailed comparisons between numerical results and theoretical predictions, clarifying the conditions under which the Landau-Zener tunneling picture holds and highlighting finite-size effects.

Weaknesses

(1) Conceptual novelty is not very high.

(2) There is room for improvement in terms of providing information and making the paper more accessible to readers by adding additional figures.

Report

In this paper, the authors study the dielectric breakdown of
one-dimensional Mott insulators, using the Fermi-Hubbard model
with a mesoscopic size.
They numerically calculate the time evolution of current
when the magnetic flux is inserted in the ring of the Hubbard chain
(i.e., the linear sweep of the Peierls phase).

First they determine the excitation gap of the system
through the Landau-Zener formalism.
The authors regard the excitation process
from the ground state to the first excited state
as the tunneling between the two level system,
and extract the excitation gap for the oscillation of current,
which agrees with the value obtained from exact diagonalization.
They also confirm that the amplitude of the oscillation of current
scales as $\log A \propto -1/E$ concerning with the applied electric field.
They further investigate the dependence of the threshold of field strength
on the gap, which is compared with the formula predicted by
the tunneling probability formula in the Landau-Zener process.
The agreement between numerics and the formula is good
for the small gap regime, but they show deviation
when the gap is large and the correlation length becomes
shorter than the system size.

While the idea that the dielectric breakdown in a Mott insulator
is described by a Landau-Zener tunneling process is already known,
and the idea of utilizing the expression for tunneling probability
arises naturally, the present work provides a systematic numerical
investigation over a wide range of parameter region.
The detailed comparison with the numerical results and prediction
by the formula helps to clarify the picture of excitation process
of Mott insulators, and the results should be of interest
to researchers of nonequilibrium phenomena in strongly correlated systems.
Thus I think that the manuscript is suitable for publication.
Before its publication, I would like the authors
to consider the following points.

Requested changes

(1) Adding the figures for the Fourier transform of current
shown in Figs. 3 and 5 would be helpful for the readers.

(2) In Fig. 8, I am interested in how the numerical data starts to
deviate from the formula as the gap increases.
I would like to ask the authors to take more data points
between the filled marker and open marker
($1.8<\Delta<2$ for $L=6$ and $1.0<\Delta<1.1$ for $L=10$).

(3) In Fig. 8, the open marker is used for $\xi<L$.
Is Eq.(10) used for the estimation of $\xi$?
If so, is it possible to evaluate $\xi$ numerically
from Eq.(10) by calculating $E_0$ while varying $\Phi$?

Recommendation

Ask for minor revision

  • validity: high
  • significance: good
  • originality: good
  • clarity: high
  • formatting: excellent
  • grammar: excellent

Author:  Joan Triadú  on 2025-05-26  [id 5520]

(in reply to Report 2 on 2025-04-20)

We want to thank the referee for the detailed report. We acknowledge that the first version of the manuscript lacked some plots that could help the readers to better understand our work. We revised it and implemented the suggested changes.

  • Adding the figures for the Fourier transform of current shown in Figs. 3 and 5 would be helpful for the readers.

Thank you for this observation. The Fourier transform of the charge current is an important element in our quantitative analysis, and it was not explicitly shown in any part of the first version of the manuscript. We use the Fourier transform of the non-equilibrium current oscillations (Figure 5) to obtain the charge gap corresponding to a certain onsite interaction. Although we refer to the Fourier transform several times in the text, the plots themselves were not included. We have added the Fourier transform of the corresponding curves in a new column of Figure 5. Additionally, including the Fourier transform in Figure 3 provides clearer insight into the shift from the Aharonov-Bohm period to the Bloch period, especially in the interacting regime.

Changes: We have added the Fourier transform of the charge current shown in Figures 3 and 5 in an additional column. We also added comments on the new plots in the main text.

  • In Fig. 8, I am interested in how the numerical data starts to deviate from the formula as the gap increases. I would like to ask the authors to take more data points between the filled marker and open marker ($1.8<\Delta<2$ for $L=6$ and $1.0<\Delta<1.1$ for $L=10$).

Thank you for the suggestion. We agree that providing more data points on the range in which the results start to diverge from the expected behavior is useful to see how the deviation happens.

Changes: We have added two additional data points in Figure 8 for each lattice size within the suggested intervals. We also added the new data points with solid markers to Figure 9.

  • In Fig. 8, the open marker is used for $\xi<L$. Is Eq.(10) used for the estimation of $\xi$? If so, is it possible to evaluate $\xi$ numerically from Eq.(10) by calculating $E_0$ while varying $\Phi$?

In Figure 8, Eq. (10) is not used for the estimation of $\xi$, but Eq. (9). The latter expression is derived in Ref. [83] and in Appendix A of Ref. [84] by combining numerical methods and the Bethe ansatz formalism. However, the first equality of Eq. (10), which relates the Drude weight to the curvature of the spectrum, is used by the authors as the starting point of their derivation. Whether could also be calculated by numerical means, we believe that this is indeed possible. Surely, it would require alternative methods beyond exact diagonalization to calculate the ground state for $L>12$. Then, a thorough study of the scaling of the spectrum curvature in terms of $U$ and $L$ should be performed. We believe that this procedure is beyond the scope of this work, but we note it as a future line of investigation.

Changes: In the caption of Figure 8 we highlight that the estimation of $\xi$ is done with Eq. (9).

---

## Round 2 · Referee Report · Anonymous (Referee 1) · 2025-6-6

Report

The authors have adequately addressed the comments raised by the referees. After the authors consider the following optional point, I recommend the revised manuscript for publication.

Requested changes

The grid in the newly added Figs.3b) and 3d) appears too coarse. To determine the peak positions accurately, could the authors increase the number of frequency points?

Recommendation

Ask for minor revision

---

## Round 2 · Referee Report · Anonymous (Referee 2) · 2025-6-12

Report

The authors have improved the manuscript
and my comments seem to be considered.
I think that the manuscript can be published.

Recommendation

Publish (meets expectations and criteria for this Journal)

---

## Round 2 · List of Changes

In order of appearance: - In Figure 2, we included the full spectrum within the displayed range. - Replaced “Aharonov period” with “Aharonov-Bohm period” on page 5. - Added the Fourier transform of the current in Figure 3. - Added a paragraph after Eq. (17) discussing the range of interaction strength and lattice sizes for which Eq. (17) is valid. - Added the Fourier transform of the current in Figure 5. - Added a sentence on the range of validity of Eq. (24) in terms of $E$ and emphasized that if $\Delta_0\approx\Delta$, Eq. (24) fails regardless of the field strength. - In Figure 8, added two data points in the range of $1.8<\Delta<2$ for $L=6$ and two more data-points in the range of $1.0<\Delta<1.1$ for $L=10$. The new data points with solid markers were also added to Figure 9. - The full (empty) markers on Figure 8 were mistakenly labeled as the $L<\xi$ ($L>\xi$ ) regime when they corresponded to $L-1<\xi$ ($L-1 >\xi$). This has been corrected. We believe that with this change, the important point is not altered: our numerical approach starts to fail when $L\approx\xi$. - In the caption of Figure 8, we now highlight that the estimation of $\xi$ is done with Eq. (9). - Added a sentence on the discussion on how the critical behavior resembles to the one of a single-particle system. - Added the Quantum Spain project in the funding information section. - Updated Ref. [54] because it has been published in a journal.

---

## Editorial Decision

published